# Identification of a Candidate Gene Set Signature for the Risk of Progression in IgM MGUS to Smoldering/Symptomatic Waldenström Macroglobulinemia (WM) by a Comparative Transcriptome Analysis of B Cells and Plasma Cells

**DOI:** 10.3390/cancers13081837

**Published:** 2021-04-12

**Authors:** Alessandra Trojani, Barbara Di Camillo, Luca Emanuele Bossi, Livia Leuzzi, Antonino Greco, Alessandra Tedeschi, Anna Maria Frustaci, Marina Deodato, Giulia Zamprogna, Alessandro Beghini, Roberto Cairoli

**Affiliations:** 1Department of Hematology, ASST Grande Ospedale Metropolitano Niguarda, 20162 Milan, Italy; lucaemanuele.bossi@ospedaleniguarda.it (L.E.B.); livia.leuzzi@ospedaleniguarda.it (L.L.); alessandra.tedeschi@ospedaleniguarda.it (A.T.); annamaria.frustaci@ospedaleniguarda.it (A.M.F.); marina.deodato@ospedaleniguarda.it (M.D.); giulia.zamprogna@ospedaleniguarda.it (G.Z.); roberto.cairoli@ospedaleniguarda.it (R.C.); 2Department of Information Engineering, University of Padova, 35131 Padova, Italy; barbara.dicamillo@unipd.it; 3Division of Hematology and Bone Marrow Transplant, Cardinale Giovanni Panico Hospital, Tricase, 73030 Lecce, Italy; agreco.dr@gmail.com; 4Department of Health Sciences, University of Milan, 20142 Milan, Italy; alessandro.beghini@unimi.it

**Keywords:** Waldenström Macroglobulinemia, IgM monoclonal gammopathies of undetermined significance, gene expression profiling, DEGs (differentially expressed genes), KEGG signaling pathways

## Abstract

**Simple Summary:**

IgM monoclonal gammopathy of undetermined significance (IgM MGUS) is an early precursor stage of the rare lymphoma Waldenström Macroglobulinemia (WM). Although comparative gene expression studies on WM, IgM MGUS, and normal B-cells (CTRLs) identified several differentially expressed genes, reliable predictors of progression from IgM MGUS to WM have not yet been identified. We performed a microarray study on CD19+ and CD138+ cells of WM vs. IgM MGUS vs. CTRLs to determine gene signatures for both cell populations. We demonstrated that hematopoietic antigens, cell-adhesion molecules, Wnt-signaling, BCR-signaling, calcium signaling, coagulation cascade, and pathways responsible for cell cycle and proliferation were significantly enriched for genes expressed in B-cells of WM vs. IgM MGUS vs. CTRLs. Interestingly, we showed nine genes which displayed the same expression levels in WM and IgM MGUS compared to CTRLs, suggesting their possible role in the risk of transformation of IgM MGUS to WM.

**Abstract:**

Waldenström Macroglobulinemia (WM) is a B-cell lymphoma characterized by the precursor condition IgM monoclonal gammopathies of undetermined significance (IgM MGUS). We performed a gene expression profiling study to compare the transcriptome signatures of bone marrow (BM) B-cells and plasma cells of 36 WM patients, 13 IgM MGUS cases, and 7 healthy subjects used as controls (CTRLs) by Affymetrix microarray. We determined 2038 differentially expressed genes (DEGs) in CD19+ cells and 29 DEGs genes in CD138+ cells, respectively. The DEGs identified in B-cells were associated with KEGG pathways, mainly involved in hematopoietic cell lineage antigens, cell adhesion/focal adhesion/transmembrane proteins, adherens junctions, Wnt-signaling pathway, BCR-signaling pathway, calcium signaling pathway, complement/coagulation cascade, platelet activation, cytokine-cytokine receptor interactions, and signaling pathways responsible for cell cycle, apoptosis, proliferation and survival. In conclusion, we showed the deregulation of groups of genes belonging to KEGG pathways in the comparison among WM vs. IgM MGUS vs. CTRLs in B-cells. Interestingly, a small set of genes in B-cells displayed a common transcriptome expression profile between WM and IgM MGUS compared to CTRLs, suggesting its possible role in the risk of transformation of IgM MGUS to WM.

## 1. Introduction

Waldenström Macroglobulinemia (WM) is a rare malignancy B-cell lymphoma characterized by a lymphoplasmacytic marrow infiltration and an uncontrolled monoclonal IgM production in the bone marrow and other organs. WM has an annual incidence of 3.4 per million among males and 1.7 among the females in United States, while an incidence of 7.3 for males and 4.2 for females occurs in Europe, respectively [1].

The disease course starts from asymptomatic WM (smoldering/smWM) form to symptomatic WM (sWM) which can manifest clinical features of lymphoadenopathy, hepatosplenomegaly, thrombocytopenia, pancytopenia, cold agglutinine syndrome, paraprotein-related cryoglobulinemia, peripheral neuropathy, or symptomatic hyperviscosity [2,3]. WM is preceded by a precursor condition, the IgM monoclonal gammopathy of undetermined significance (IgM MGUS), which is an asymptomatic form characterized by a serum IgM monoclonal protein <3 g/dL, and a bone marrow lymphoplasmacytic infiltration below 10% [4]. IgM MGUS has an overall risk of progression of 1.5–2% per year, and approximately 18% at 10 years to smWM or other lymphoproliferative disorders. While patients with smWM show a probability of progression to WM, amyloidosis or lymphoma of 12% per year, and 65% at 10 years [5,6]. Recurring somatic mutations in *MYD88, CXCR4, ARID1A*, and *CD79B* have been found in bone marrow lymphoplasmacytic cells by next-generation sequencing (NGS) in patients with WM [7,8,9,10]. MYD88 L265P mutation has been found in nearly 90% of WM patients and in 47% of cases with IgM MGUS. SmWM patients with wild-type MYD88 have a higher risk to develop symptomatic lymphoma and display poor response to treatment and shorter overall survival [1,7,11]. A previous study demonstrated that IgM MGUS subjects with MYD88 L265P mutation have a higher risk of progression to WM or other lymphoproliferative diseases and a higher disease burden, indicating *MYD88* gene as an important oncogenic driver [12,13]. *CXCR4* somatic mutations have been observed in more than 40% of WM patients. They do not adversely impact overall survival but play a role in guiding treatment [7]. The deletion of chromosome 6q (del6q) occurs in about 50% of patients with WM and is associated with shorter survival [14]. The complexity of WM clones harbors clonal B lymphocytes, lymphoplasmacytic cells, and plasma cells secreting a monoclonal immunoglobulin M (IgM).

The results provided by gene expression profiling studies (GEP) highlighted differentially expressed genes (DEGs) involved in oncogenesis and B-cell differentiation in the comparison between B-cells and plasma cells of WM vs. multiple myeloma (MM) vs. chronic lymphocytic leukemia (CLL) counterparts, respectively [15]. A comparative gene expression analysis between WM, CLL, and MM showed that the over expression of *IL6* and MAPK signaling pathway were unique to WM [16].

We previously determined the up regulation of JAK/STAT, PI3K/Akt/mTOR, and MAPK signaling pathways between WM and IgM MGUS CD19+ cells whereas immune response and cell activation mainly distinguished WM from IgM MGUS CD138+ cells [17].

A careful multiparametric flow cytometry analysis showed a strong similarity of immunophenotypic expression profile between clonal B-cells of IgM MGUS, smWM, and sWM [18].

A gene expression and mutational study demonstrated that genes involved in the Toll-like receptors (TLR) signaling pathways were up-regulated in symptomatic WM vs. indolent forms [5]. Moreover, the authors demonstrated a higher incidence of gene mutations during the process of transition from IgM MGUS to smWM, and sWM. Interestingly, an extensive study demonstrated a higher risk of progression to WM and a lower overall survival in subjects with IgM MGUS compared to a matched control population [6].

Despite the improvement in gene expression signatures and genomic alterations responsible for WM, trustworthy predictors of progression from IgM MGUS to WM have not yet been identified.

In the current study, we selected B-cells and plasma cells of bone marrow of 36 WM patients, 13 IgM MGUS cases, and 7 healthy subjects (CTRLs). Therefore, we performed a wide-transcriptome microarray analysis to determine statistically significant gene expression differences among WM vs. IgM MGUS vs. CTRLs in CD19+ cells and CD138+ cells, respectively. We also investigated the differentially expressed genes using gene ontology (GO) and Kyoto Encyclopedia of Genes and Genomes (KEGG) to determine candidate genes for the risk of progression of IgM MGUS to WM. Our GEP data identified different patterns of significant DEGs in B-cells (*n* = 2038), while we found only 29 significant DEGs in the comparison among plasma cells of WM vs. IgM MGUS vs. CTRLs.

B-cells showed significant gene expression differences in WM vs. IgM MGUS vs. CTRLs in hematopoietic cell lineage markers, transmembrane/cell adhesion molecules, focal adhesion, adherens junctions, B-cell Receptor (BCR) signaling pathway, complement coagulation cascade and platelet activation, cytokine-cytokine receptor interactions/IL-17 signaling pathway, WNT/β catenin signaling pathway, Calcium signaling pathway, JAK/STAT signaling pathway, FOXO signaling pathway, NFKB signaling pathway, and p53 signaling pathway. Finally, we identified nine genes which showed similar expression levels in WM and IgM MGUS with respect to CTRLs, suggesting a candidate gene set signature predictive of the risk of progression of IgM MGUS to WM.

## 2. Results

### 2.1. Total RNA Concentration in BM CD19+ Cells and BM CD138+ Cells

The total RNA concentration isolated from 1 million cells (CD19+ or CD138+ cells) was about 1–2 µg and it was measured by Nanodrop (Thermo Fisher Scientific, Milan, Italy).

### 2.2. GEP Results

From the comparison among WM, IgM MGUS, and CRTLs of CD19+ cells, we identified 3052 probe-sets corresponding to 2038 unique genes and 584 not annotated probe-sets.

GEP analyses among WM, IgM MGUS and CRTLs of CD138+ cells determined 38 probe-sets corresponding to 29 unique genes and six not annotated probe-sets. There were 16 genes in common between the CD19+ and CD138+ cells.

To better characterize the genes selected as differentially expressed, we considered, for all genes that resulted significantly different expressed in at least one of the below listed comparisons, the following patterns of differential expression: p1) WM > IgM MGUS and IgM MGUS > CTRL, p2) WM < IgM MGUS and IgM MGUS < CTRL, p3) WM > IgM MGUS but IgM MGUS < CTRL, p4) WM < IgM MGUS but IgM MGUS > CTRL. We performed a functional annotation clustering and enrichment analysis on each pattern using DAVID 6.8 (https://david.ncifcrf.gov/, 24 July 2020). Complete results are reported in Appendix A.

### 2.3. CD19+ Patterns

For CD19+ cells, we found that patterns p1, p2, p3, and p4 contained 265 unique genes (377 probe-sets of which 58 not annotated), 403 unique genes (683 probe-sets of which 135 not annotated), 827 unique genes (1211 probe-sets of which 218 not annotated), and 543 unique genes (781 probe-sets of which 173 not annotated), respectively.

#### 2.3.1. Transmembrane Proteins

Functional clustering has highlighted genes encoding transmembrane proteins (*n* = 20). Among the DEGs, all the genes (encoding transmembrane proteins) shown in Appendix A were over expressed in WM vs. IgM MGUS and vs. CTRLs. No significant gene expression changes have been noted between IgM MGUS and CTRLs (except for *PTPRG*). These findings suggest that the deregulation of these genes was specific for WM B-cells.

#### 2.3.2. Hematopoietic Cell Lineage (KEGG Pathway hsa04640)

GEP data highlighted eight genes encoding surface hematopoietic cell lineage markers which could determine different cellular stages (Appendix A). Data showed that *IL2RA* (CD25)*, CD1D*, and *CD44* were over expressed in WM vs. IgM MGUS and vs. CTRLs (p1). *CD4* (helper T cells) was up regulated in IgM MGUS vs. WM and vs. CTRLs (p4).

#### 2.3.3. Cell Adhesion Molecules (CAMs) (KEGG Pathway hsa04514)

The functional clustering analysis highlighted several genes encoding CAMs (selectins, cadherins, integrins, immunoglobulins), which are expressed on the surface of B lymphocytes and are involved in the binding with other cells and, in particular, with immune system (T cells), neural system (pre- and post-synaptic neurons), and inflammation processes. The Human leukocyte antigen (HLA) system is a group of proteins that are encoded by the major histocompatibility complex (MHC) genes such as *HLA-A, HLA-C, HLA-B, HLA-E*, and *HLA-G* which were over expressed in WM vs. IgM MGUS and vs. CTRLs (Appendix A). None of these genes were significantly different expressed between IgM MGUS and CTRLs. In addition, we demonstrated that *IGHM* (Constant region of the mu heavy chain of IgM), which defines the IgM isotype, was over expressed in WM vs. IgM MGUS and vs. CTRLs, while no differences in expression were noted between IgM MGUS and CTRLs (Appendix A).

Our results showed that *CD86* (B-cell surface molecule) which regulates T cell responses, was down-regulated in WM vs. IgM MGUS with the fold change (FC = −2.93) and in WM vs. CTRL (FC = −2.02) (Appendix A).

We found that *CD58* (plasma membrane protein) and *CEACAM1* (CD66a) (angiogenesis) were up regulated in WM vs. IgM MGUS and vs. CTRLs as shown in Appendix A.

Notably, *CDHR3* (calcium-dependent cell adhesion proteins) was over expressed in CTRLs vs. IgM MGUS and vs. WM while no significant expression changes were observed in the comparison between WM and IgM MGUS (Appendix A).

*CNTNAP2* was down regulated in WM vs. IgM MGUS and vs. CTRLs. Notably, *ADAM23* was down regulated in WM vs. CTRLs and in IgM MGUS vs. CTRLs, but no significant differences in expression were found between WM and IgM MGUS (Appendix A).

#### 2.3.4. Focal Adhesion (KEGG Pathway hsa04510)

We have demonstrated an altered expression of 11 genes which regulate focal adhesion pathway in B-cells (Appendix A). Focal adhesion molecules are assigned to cell motility, proliferation, and survival by the regulation of acting cytoskeleton, Wnt signaling pathway, and PI3K/MAPK signaling pathway. *ITBG5* and *ITGA2B* were under expressed in WM vs. IgM MGUS whereas *ITGB3, PDGFA* (Platelet Derived Growth Factor Subunit A), *CAV1* (Caveolin 1)*, PTK2* (encoding FAK), and *IGF1R* (insulin-like growth factor 1 receptor) were under expressed in WM vs. IgM MGUS and vs. CTRLs. On the other hand, *RAPGEF1* was over expressed in WM vs. IgM MGUS and vs. CTRLs.

#### 2.3.5. Adherents Junctions (KEGG Pathway hsa04520) and WNT/β-Catenin Signaling Pathway (KEGG Pathway hsa04310)

Our GEP data identified some DEGs in adherens junction KEGG pathway (Appendix A). Adherens junction proteins permit the intercellular adhesion and influence cell polarity, movement, and proliferation. We showed the over expression of *PTPRJ* and *PTPN1* in WM vs. IgM MGUS and vs. CTRLs. Both genes encode tyrosine phosphatases which dephosphorilate β-catenin, thus affecting cell-cell adhesion, cell migration, and β-catenin levels. In addition, *SSX2IP* and *TJP1* which regulate the organization of the actin cytoskeleton and the cell migration were down regulated in WM vs. IgM MGUS and vs. CTRLs. The deregulation of expression of both genes was unique to WM B-cells as no differences of expression were found between IgM MGUS and CTRLs.

Our data showed that in the canonical WNT signaling pathway, *WNT3* was over expressed in WM vs. IgM MGUS (FC = 2.19) and vs. CTRLs (FC = 2.35) (Appendix A). *LEF1*, a transcription factor involved in WNT/Beta-catenin pathway, was up regulated in CTRLs vs. WM (FC = 3.86) and vs. IgM MGUS (FC = 2.58) but no significant gene expression differences were noted between WM and IgM MGUS (Appendix A).

Within Wnt/Ca^2+^ pathway, we found 4 DEGs. Among them, *RYK* was up regulated in WM vs. IgM MGUS and vs. CTRLs (Appendix A). In addition, *PRKCA* (Protein kinase C-alpha) and *MARCKS* (myristoylated alanine-rich protein kinase C substrate) were DEGs in B-cells (Appendix A).

#### 2.3.6. Calcium Signaling Pathway (KEGG Pathway hsa04020)

We revealed the deregulation of 11 genes associated with Calcium signaling pathway which mediate the calcium entry into the cells from the outside through plasma membrane by channels (Appendix A). We have found that *ADRB2* (transmembrane Beta-adrenergic receptor) was over expressed in WM vs. CTRLs (FC = 2.31) and in IgM MGUS vs. CTRLs (FC = 1.8), but no significant expression differences were observed between WM and IgM MGUS B-cells (Appendix A). We have demonstrated that *CACNAD1*, encoding the membrane voltage-sensitive calcium channel, was up regulated in WM vs. IgM MGUS (FC = 1.79) and vs. CTRLs (FC = 2.64).

We determined the down regulation of *TRPC1* in WM vs. IgM MGUS and vs. CTRLs. *TRPC1* encodes a membrane protein that can form a non-selective channel permeable to calcium which interacts with ITPR1 and ITPR2 (https://string-db.org, 24 July 2020). *ITPR1*, which mediates the calcium release inside the cells, was under expressed in WM vs. IgM MGUS and vs. CTRLs (Appendix A). Finally, *MYLK* (muscle contraction) was under expressed in WM vs. IgM MGUS and vs. CTRLs (Appendix A).

#### 2.3.7. BCR Signaling Pathway (KEGG Pathway hsa04662)

GEP analyses have highlighted 7 DEGs belonging to BCR signaling pathway (Appendix A). *SOS1, CD79A*, and *CBLB* were up regulated in WM vs. IgM MGUS and vs. CTRLs, whereas *SYK* was over expressed in WM vs. IgM MGUS, but no expression changes were noted between WM and IgM MGUS vs. CTRLs, respectively (Appendix A). *RASGRP3* was under expressed in WM vs. CTRLs as well as in IgM MGUS vs. CTRLs but it was not differentially expressed between WM and IgM MGUS (Appendix A). We found that *MAP4K1* (MAPK signaling pathway), which may play a role in hematopoietic lineage decisions and growth regulation, was over expressed in WM vs. IgM MGUS and vs. CTRLs. Finally, we demonstrated the down regulation of *PIK3AP1* in WM and IgM MGUS vs. CTRLs B-cells, but no significant gene expression differences have been found between WM and IgM MGUS (Appendix A).

#### 2.3.8. Complement and Coagulation Cascade (KEGG Pathway hsa04610) and Platelet Activation (KEGG Pathway hsa04611)

*VWF* (von Willebrand factor), *TFPI*, and *SERPINE1* belonging to the complement and coagulation cascade (hsa04610), were under expressed in WM vs. IgM MGUS but no expression differences were observed between CTRLs and WM as well as CTRLs and IgM MGUS (Appendix A).

Within the platelet activation pathway, we revealed the deregulation of *PTGIR*, *P2RY12* both encoding membrane receptors, and *RAP1A* which plays a role in the platelet activation by controlling the cell-cell and cell-matrix interactions (Appendix A).

#### 2.3.9. Cytokine-Cytokine Receptor Interactions (KEGG Pathway hsa04060) and IL-17 Signaling Pathway (KEGG Pathway hsa04657)

We observed that among cytokine-cytokine receptor interactions pathway, *PF4, PF4V1*, and *PPBP* were down regulated in WM vs. IgM MGUS and vs. CTRLs with very high FC (Appendix A).

Among the IL17-like cytokines (KEGG hsa04657), our data showed the over expression of *IL17RB* (interleukin 17 receptor B) in WM vs. IgM MGUS and vs. CTRLs with high FC (Appendix A).

#### 2.3.10. JAK/STAT Signaling Pathway (KEGG Pathway hsa04630)

Our GEP data highlighted the deregulation of six crucial genes involved in the JAK/STAT signaling pathway (Appendix A). The activation of JAK/STAT signaling pathway in WM could promote the over expression of genes involved in proliferation and differentiation. We demonstrated that *BCL2, SOS1*, and *STAT3* were over expressed in WM vs. IgM MGUS and vs. CTRLs, but no gene expression changes were noted between CTRLs and IgM MGUS.

In addition, our results have shown that *IL4R*, *IL13RA1*, and *IL21R* were down regulated in WM vs. IgM MGUS and vs. CTRL with high FC (Appendix A), but similar expression changes occurred between CTRLs and IgM MGUS. Based on our GEP data, we suggest that the deregulation of the JAK-STAT signaling pathway was specific for WM B-cells.

#### 2.3.11. FOXO Signaling Pathway (KEGG Pathway hsa04068) and TGF-β Signaling Pathway (KEGG Pathway hsa04350)

We revealed 8 DEGs in FOXO signaling pathway which regulates proteins involved in apoptosis, cell-cycle control, and the hematopoietic stem cell self-renewal (Appendix A). *FOXO3, IL6*, and *CDKN1B* were over expressed in WM vs. IgM MGUS and vs. CTRLs, but no expression differences were observed between IgM MGUS and CTRLs in all these genes. These results suggest that these genes were uniquely deregulated in WM B-cells.

To be observed is the fact that *IL6* was over expressed in WM vs. IgM MGUS (FC = 2.65) and in WM vs. CTRLs (FC = 2.91). Interestingly, *CDKN1B* was up regulated in WM vs. IgM MGUS (FC = 1.5) and vs. CTRLs (FC = 1.5), demonstrating the same expression level in IgM MGUS and CTRLs. Our data identified the progressive over expression of *RAG1* in WM vs. IgM MGUS vs. CTRLs with high FC (Appendix A). *TGFB2* and *SMAD3* were under expressed in WM vs. IgM MGUS and vs. CTRL. In particular, *TGFB2* displayed the same expression level in IgM MGUS and CTRLs (Appendix A).

*SMAD6* was up regulated in IgM MGUS vs. WM and vs. CTRLs. Finally, *EP300* was up regulated in WM vs. IgM MGUS and down regulated in IgM MGUS vs. CTRLs but no significant expression difference was noted between WM and CTRLs. We will further investigate the possible role of this gene in IgM MGUS subjects.

#### 2.3.12. NFKB Signaling Pathway (KEGG Pathway hsa04064)

Our results demonstrated the over expression of *NFKBIZ, RIPK1, CRKL*, and *CFLAR* in WM vs. IgM MGUS (p3) (Appendix A). To be noted is the fact that *NFKBIZ* has shown the same expression level in IgM MGUS and CTRLs, suggesting that its expression was uniquely altered in WM. *RIPK1* and *CRKL* were under expressed in IgM MGUS vs. CTRLs, but no differences were seen between WM and CTRLs. As far as we know, both genes have not yet been identified in WM and we will evaluate their possible role in IgM MGUS. Ultimately, our findings have highlighted the over expression of the apoptotic suppressor gene *XIAP* in WM vs. IgM MGUS and vs. CTRLs in B-cells. From our data, we could hypothesize that the NFKB signaling pathway (apart from *CRKL*) deregulation occurred only in WM B-cells.

#### 2.3.13. p53 Signaling Pathway (KEGG Pathway hsa04115)

We have demonstrated that *TP73*, *CD82*, and *CHEK1* involved in p53 signaling pathway, were DEGs in B-cells (Appendix A). *TP73* was progressively over expressed from CTRLs to IgM MGUS and WM, whereas CD82 was up regulated in WM vs. IgM MGUS and vs. CTRLs, but no significant differences of expression were evaluated between IgM MGUS and CTRLs. Interestingly, we showed that *CHEK1* was down regulated in WM vs. CTRLs and in IgM MGUS vs. CTRLs, but no expression differences were observed between WM and IgM MGUS (Appendix A).

#### 2.3.14. Zinc Finger Proteins

Zinc finger proteins (ZNF) (*n* = 120) were deregulated in all the 4 patterns CD19+. We suggest the possible role of ZNF in the transcription regulation and cancer progression in the comparison among the three groups of subjects. Additional studies are needed to confirm these data. Among them, we focused on *ZNF804A* and *ZNF215* which were up regulated in WM vs. IgMGUS and vs. CTRLs with very high FC, while *IKZF2* was progressively up regulated from WM to IgM MGUS and CTRLs (Table 1).

#### 2.3.15. Genes Belonging to Pattern 2 with Very High FC in CD19+ Cells

Herein, we show the deregulation of some genes which have displayed very high FC in the p2 CD19+ cells (Table 2). We identified the under expression of *ADAM28* in WM vs. IgM MGUS and vs. CTRLs.

*ADARB1* and *APBB2* were progressively over expressed from WM to IgM MGUS to CTRLs. Interestingly, *EZH2* and *HIST1H1B* were over expressed in CTRLs vs. WM and vs. IgM MGUS, but no expression changes were evaluated between WM and IgM MGUS (Table 2). The role of both genes in WM has not been described yet, and future investigations are needed. We report that *EZH2* mutations have been described in follicular and large B-cell lymphomas (DLBCL) [19].

#### 2.3.16. Candidate Genes of the Risk of Progression of IgM MGUS in WM

Our GEP results highlighted nine genes which displayed a similar expression level between WM and IgM MGUS compared to CTRLs in B-cells, suggesting their possible role in the risk of transformation from IgM MGUS to WM (Table 3).

To better appreciate the trend of expression of these genes, Figure 1 shows their boxplot in WM, IgM MGUS and CTRL groups.

Clustering and principal component analysis applied to the analyzed subjects, considering only the expression signature of the nine candidate genes of the risk of progression of IgM MGUS in WM, highlighted a quite homogeneous cluster of CTRL subjects, whereas WM and IgM MGUS subjects tended to group together. This is consistent with the observation that these nine genes displayed the same expression levels in both WM and IgM MGUS compared to CTRLs (Table 3). To further inspect the similarities and the differences among WM and IgM MGUS subjects, we computed the Euclidean pair-wise distance between subjects and, using this distance as weight, constructed a minimum spanning tree (MST) (see Section 4.) showing, within the same graph, the subjects as colored nodes and the pairs of closest subjects connected by edges (Figure 2). Blue, green and red dots indicate CTRL, IgM MGUS and WM samples, respectively. Figure 2 clearly shows that, whereas CTRL subjects cluster together, IgM MGUS subjects lay in different paths between CTRL and WM subjects.

### 2.4. CD138+ Cells

From the comparison between WM vs. IgM MGUS vs. CTRLs in the CD138 cells, we found 26 genes deregulated in pattern 2 with high FC ranging up to 6.2, while there were 0, 1 and 2 unique genes in patterns 1, 3 and 4, respectively (Table 4).

#### 2.4.1. Adherens Junction (KEGG Pathway hsa04520), Gap Junction (KEGG Pathway hsa04540), and Tight Junction (KEGG Pathway hsa04530) in Pattern 2 CD138+ Cells

Functional clustering performed in p2 has shown that *ARHGAP32, TJP1* (KEGG hsa04520, hsa04540, hsa04530)*, PARD3* (KEGG hsa04520, hsa04530), *CADPS2*, and *SYNM* encoding cell junction proteins, were down regulated in WM vs. IgM MGUS and vs. CTRLs with very high FC (Table 4). In particular, *TJP1* and *PARD3* were involved in tight junction as well as adherens junctions, and both control cell polarity and junction assembly (Table 4). To be observed is the fact that *TUBB2B* encodes tubulin which is the major constituent of microtubules and takes part also in the gap junction pathway. This gene was over expressed in CTRLs vs. WM (FC = 3.03) and vs. IgM MGUS (FC = 2.92), but no expression differences were observed between WM and IgM MGUS (Table 4). Thus, we could suppose that *TUBB2B* might play a role in the risk of transformation of IgM MGUS to WM in plasma cells, but further studies will be needed to clarify these results.

#### 2.4.2. Cell Adhesion Molecules (KEGG Pathway hsa04514) in Pattern 2 CD138+ Cells

*CNTN1*, which encodes a CAM and mediates cell-cell adhesion in the nervous system, was down regulated in WM vs. IgM MGUS and vs. CTRLs (Table 4).

#### 2.4.3. p53 Signaling Pathway (KEGG Pathway hsa04115) in Pattern 2 CD138+ Cells

We demonstrated that *PERP* (p53 apoptosis effector) was progressively up regulated from WM to IgM MGUS and to CTRLs with high FC, respectively (Table 4).

#### 2.4.4. Calcium Signaling Pathway (KEGG Pathway hsa04020) in Pattern 2 CD138+ Cell

MYLK was down regulated in WM vs. IgM MGUS and vs. CTRLs, as already observed in CD19+ cells (Appendix A).

### 2.5. Common Genes Deregulated in CD19+ Cells and CD138+ Cells

GEP data identified 15 genes (of which four are not annotated) in common between CD19+ cells and CD138+ patterns shown in Table 5.

In particular, *LAPTM4B*, *PLA2G2D*, *KIAA1804*, *MYLK*, *ARHGAP32*, *SYNM*, and *MARC2* clustered in p2 CD19+ cells as well as in p2 CD138+ cells. *TJP1*, *PPARGC1A*, *GBA3*, and *NFIB* were down regulated in WM p4 CD19+ and in WM p2 CD138+ cells, respectively. However, these genes were significantly under expressed in WM vs. IgM MGUS.

## 3. Discussion

The current state of the art for genomics and transcriptome knowledge of WM points out the complexity of this disease. The significant degree of clinical and genetic heterogeneity can be attributed mostly to WM tumoral clone compartment which consists of clonal B-cells, clonal plasma cells and a lymphoplasmacytic population [7,20]. Whole-genome sequencing studies have shown recurrent somatic mutations in WM and IgM MGUS as the most important prognostic markers defining prognosis, risk of progression, and treatment decisions [21]. Previous GEP profiling studies have identified several pathways and gene expression alterations in WM and IgM MGUS patients [15,16]. Despite transcriptome investigations, little is known about the pathways and the gene expression changes that lead IgM MGUS to progress to WM disease.

To the best of our knowledge, we report an extensive transcriptome gene expression comparison of B-cells and plasma cells of WM patients vs. IgM MGUS subjects vs. healthy donors in order to identify possible genes and pathways involved in the risk of progression from IgM MGUS to WM. We distinguished IgM MGUS from WM based on the bone marrow infiltrate following the consensus panel recommendations from the second IWWM as described in Section 4.

We have chosen to perform a comparison among the three groups of subjects to investigate an evolutionary pattern associated with transformation. A possible limitation of our approach is that the selected CD19+ cell population of IgM MGUS subjects might include regular non-clonal CD19+ cells which could contribute to the GEP signals. To partially take this eventuality into account, we chose to compare IgM MGUS CD19+ cells vs. normal CD19+ cells to include the gene deregulation referred to normal condition, in order to evaluate the gene expression changes due to the presence of the clonal CD19+ cells.

Another possible limitation of our study is that, at the time of samples collection, the *MYD88* and *CXCR4* mutation status were not monitored, whereas current literature reports an extreme degree of heterogeneity observed in WM based on *MYD88* and *CXCR4* mutational status [7,8,9,10,11,12,13]. However, despite this possibly high variance, statistical tests were able to identify a significant difference after correction for multiple testing. Our GEP data showed that 2038 unique genes were significantly differentially expressed in the comparison between WM vs. IgM MGUS vs. CTRLs in CD19+ cells. Among them, we focused on 115 DEGs which were grouped in KEGG pathways and we investigated their possible role in the biology of WM. Some of the deregulated genes were previously implicated in WM, as shown hereafter.

On the contrary, only very few genes, which encoded adherens and gap junction proteins, cell adhesion molecules, p53 signaling pathway and calcium signaling pathway, came from the analysis of CD138+ cells of WM vs. IgM MGUS vs. CTRLs.

Herein, we identified a small gene set (*n* = 9) which displayed no significant differences of expression between WM and IgM MGUS B-cells with respect to CTRLs (Table 3, Figure 1 and Figure 2). Thus, we suggest that these genes could be involved in the biological processes of transformation of IgM MGUS in WM. Until now, the 13 IgM MGUS subjects have not been transformed in WM or other NHL. They have been monitored in follow-up every 6 months. The observation of the IgM MGUS cases and their possible transformation to lymphoma will provide a new insight in our findings. Our next study will investigate the expression levels of the nine genes in a larger cohort of IgM MGUS cases. The gene expression signature will thus be investigated in new diagnosed patients, and after progression in WM. Further studies of the role of the nine genes in the biology of WM will be performed in a larger cohort of patients. The determination of the biological causes of WM transformation could help to corroborate the risk factors of disease progression from IgM MGUS to WM, and to recognize who is at higher risk and need closer follow-up.

### 3.1. Different Expressions of Genes Encoding Transmembrane Molecules, Hematopoietic Cell Surface Antigens, Cell and Focal Adhesion Proteins in B Cells

Our GEP results demonstrated that transmembrane proteins on B-cells were encoded by DEGs. Among them, *CNR1* was up regulated in WM with respect to IgM MGUS and CTRLs. A previous study showed that *CNR1* was over expressed in Mantle Cell Lymphoma (MCL) B-cells compared to non-malignant B-cells and that targeting the endocannabinoid receptors was suggested in the treatment of MCL [22].

Some studies extensively investigated the immunophenotypic features of WM and IgM MGUS B-cells and plasma cells [18,23,24]. Herein, GEP data highlighted genes encoding hematopoietic surface antigens as differentially expressed in B-cells of WM vs. IgM MGUS vs. CTRLs. The over expression of *IL2RA* in B-cells showed by our study is strongly supported by previous authors who have demonstrated that Waldenström’s clone was characterized by CD25 marker [18]. In addition, *CD1D* could play a role as a diagnostic marker in the characterization of B-cell chronic lymphoproliferative disorders (B-CLPDs) [25]. We demonstrated that *CD1D* expression on B-cells was higher in WM vs. IgM MGUS and vs. CTRLs, suggesting its possible role as a marker in WM.

We found that *CD44*, which encodes a glycoprotein over expressed in cancer stem cells, was up regulated in WM B-cells [26]. Some authors determined that T-cell markers were over expressed in a little subset of WM B-cells, probably due to a re-differentiation during tumor development [27]. We showed that *CD4* (T-Cell Surface Glycoprotein) was up regulated in IgM MGUS vs. WM and in WM vs. CTRLs B-cells suggesting the presence of this protein on B-cell surface. Its role needs to be further investigated.

Among the CAMs, some authors identified the higher serum levels HLA-Gs in WM and IgM MGUS compared to CTRLs [28]. In addition, HLA-G is a marker of clonality since high serum levels of HLA-Gs were observed in both IgG and IgA MGUS and MM patients. We identified the over expression of some HLAs genes in WM vs. IgM MGUS and vs. CTRLs, suggesting that the higher expression of HLAs was uniquely deregulated in WM B-cells.

A previous study on trogocytosis in WM, CLL and MM demonstrated that cells can transfer surface molecules during cell–cell interactions, which can lead to the creation of new cell types with functional changes, as shown in a previous paper [29]. The authors investigated trogocytosis in patients with MM, WM and CLL, and found that CD86 and HLA-G molecules could be transferred mainly to T cells which are the best recipient cells. In addition, MM malignant plasma cells can donate CD86 and HLA-G to T cells, which can inhibit the proliferation of other T cells. Future studies are needed to clarify the role of trogocytosis as a mechanism of tumor-induces suppression of the immune system. Moreover, CD86 expression correlated with poor prognosis in patients with MM because of its prosurvival activity in malignant plasma cells [30]. In addition, an immunophenotyping study on WM cell lines has shown different expression of cell surface antigens and CD86 appeared on the surface of human WM cell lines [31]. On the contrary, our GEP data demonstrated that *CD86* was significantly up regulated in IgM MGUS and CTRLs vs. WM B-cells. Future studies are required to explore the impact of CD86 expression in B-cells of WM and IgM MGUS patients.

We have demonstrated the over expression of *CEACAM1* in WM vs. IgM MGUS and vs. CTRLs B-cells, confirming previous investigations which showed the up regulation of CD66a in bone marrow samples of non-Hodgkin lymphoma (NHL) and MM [32].

As shown in KEGG pathway hsa04514, *CNTNAP2* corroborates cell adhesion between Schwann cells and oligodendrocytes. Notably, *CNTNAP2* encodes CASPR2 protein which is expressed in central nervous system (CNS) and peripheral nervous system (PNS) playing a role in neurodevelopmental disorders and in the autoimmune disease Sjogren’s syndrome (SS). In addition, previous data showed that CASPR2 and ADAM23 are active in the peripheral neuropathy by controlling the activity of potassium channels in SS [33]. To the best of our knowledge, the role of *ADAM23* in WM has not yet been found, but the inactivation of this gene is associated with tumorigenesis in human cancers [34]. We found that *CNTNAP2* and *ADAM23* were deregulated in B-cells of both WM and IgM MGUS. Thus, according to the data shown above, we could speculate the possible role of both genes in WM patients with Sjogren’s syndrome. Based on the possible role of *ADAM23* in the peripheral neuropathy, its involvement in IgM MGUS could be supported by some studies which have investigated a large cohort of MGUS patients and found that sensory/motor neuropathies were associated with IgM MGUS patients [35,36].

Herein, our results have shown the *ADAM23* up regulation in CTRLs vs. WM and IgM MGUS, but no expression changes between WM and IgM MGUS compared to CTRLs. For this reason, we could assume a possible implication of this gene as a risk of progression from IgM MGUS to WM. However, the deregulation of *ADAM23* has a good chance to be implicated in clinical neuropathy presentations in WM as well as in IgM MGUS.

Moreover, we did not find any expression differences in *CDHR3* between WM and IgM MGUS; thus, we could suggest its possible role in the risk of progression of IgM MGUS to WM, and even its function still needs to be determined.

Among the CAMs, we previously mentioned the integrins, which also take part into the focal adhesion pathway. Our GEP data identified some DEGs (*ITGB5, ITGA2B, ITGB3, THBS1, ACTN1, CAV1, IGF1R, RAPGEF1, VAV3, PTK2, PDGFA*) in this pathway which play a role in cell motility (regulation of actin cytoscheleton), differentiation, cell proliferation, and cell survival. In particular, the interaction between membrane receptors (ITGA2B, ITGB3 and ITGB5) with THBS1, IGF1R with PDGFA, and the plasma membrane protein CAV1 with integrins, activate the Focal adhesion kinase FAK, Src and Shc proteins, to initiate signaling events for the reorganization of the actin cytoskeleton inside the cells (KEGG pathway hsa04510).

FAK has been largely investigated in solid tumors as well as in hematological diseases [37]. Indeed, some authors showed that FAK expression was present in B-lymphoblastic leukemias/lymphoma whereas it was absent in T-lymphoblastic leukemias/lymphoma and in almost all investigated myeloma cases [38]. Interestingly, a previous study has described the role of CAMs in migration and adhesion of WM cells to the bone marrow niche, including FAK [39].

### 3.2. Adherens Junctions and WNT Signaling Pathway

The Adherens junction pathway regulates the actin cytoscheleton and it is involved in the canonical Wnt signaling pathway. Within adherens junction pathway, we demonstrated the deregulation of some DEGs and, in particular, the deregulation of *SSX2IP* and *TJP1* in WM B-cells compared to IgM MGUS and CTRL B-cells. A previous GEP study demonstrated an altered *SSX2IP* expression in patients with acute myeloid leukemia based on cytogenetic translocations [40].

Several studies have demonstrated the deregulation of genes associated with the Wnt pathway in B-cell disorders and MM. Indeed, the transcriptional factor *LEF1* was under expressed in WM and IgM MGUS vs. CTRLs as well as vs. CLL B-cells [5,15,41,42]. Moreover, *LEF1* was over expressed in immature normal B-cells compared to mature normal B-cells [42]. In line with these studies, we showed that *LEF1* was down regulated in WM and IgM MGUS compared to CTRLs B-cells. We could speculate that *LEF1* could be involved in the risk of progression from IgM MGUS to WM as no significant expression differences between WM and IgM MMGUS CD19+ cells were noted (Table 3). In addition, *PRKCA*, *CAMK2G*, and *PPP3CA* were deregulated in B-cells.

Finally, our GEP data demonstrated the up regulation of *MARCKS* in WM vs. IgM MGUS and vs. CTRLs confirmed by Gutierrez et al. who demonstrated that *MARCKS* was up-regulated in WM B lymphocytes with respect to CLL B-cells [15].

### 3.3. Deregulation of Calcium Signaling Pathway

Previous studies investigated the role of genes involved in the calcium signaling pathway in lymphomas and leukemias as indicated below. The entry of calcium in the cells is due to the presence of an electrochemical gradient across the plasma membrane. A microarray study demonstrated that *ADRB2* expression was higher in MCL cell lines and in DLBCL lymphocytes compared to normal B-cells [43,44]. In addition, *CACNA1D* which mediates the entry of calcium into the cells was up regulated in CML leukemic stem cells (LSCs) compared to normal LSCs [45].

Our GEP data highlighted the deregulation of 11 genes of calcium signaling pathway in all four patterns CD19. Interestingly, *ADRB2* did not show a significant expression difference between WM and IgM MGUS and we could suppose that it might be involved in the progression risk from IgM MGUS to WM (Table 3).

Recent data showed the possible role of transient receptor potential canonical (*TRPC*) genes by performing RNAseq analysis in mice that lack the *TRPC* [46]. The authors pointed out the deregulation of *TRPC* belonging to several metabolic pathways such as PI3K-B signaling pathway, cytokine-cytokine receptor interaction, calcium homeostasis, extracellular matrix -receptor, and circadian rhythms in several human diseases. Interestingly, we found a down regulation of *TRPC1* in WM vs. IgM MGUS and vs. CTRLs. Thus, we could speculate that the deregulation of *TRPC1* gene could play a role in WM. Nevertheless, this needs to be confirmed in future investigations.

### 3.4. BCR Signaling Pathway

Previous data highlighted that B-cell receptor signaling pathway is chronically active in WM cells and can distinct WM patients’ subgroups [47].

The BCR signaling pathway is an integral protein transmembrane complex of B-cells which regulates Ig production, immune response, and autoimmunity and B-cell ontogeny. Antigens link to the BCR antigen receptors such as CD79A, activate the protein tyrosine kinase SYK and a cascade of signals [48,49]. SYK phosphorilates PIK3AP1 that in turn activates the PI3K-Akt signaling pathway. On the other hand, the cellular SYK protein can stimulate through a chain reaction mechanism, *SOS1, RASGRP3*, and *MAP4K1* that activate MAPK signaling pathway (KEGG pathway hsa04662).

Jiménez et al. demonstrated that mutations in CD79A/CD79B could represent a risk of transformation in some WM patients to DLBCL [50]. In addition, a recent study demonstrated the role of *SYK* to promote the survival of MYD88-mutated B-cell lymphoma cells in vitro [51]. This finding suggests the use of SYK inhibitors in combination with Bruton tyrosine kinase (BTK) inhibitor in MYD-88-mutated lymphomas. A genomic profiling study has demonstrated *PIK3AP1* deletions in acute lymphoblastic leukemia cells [52]. Our results underlined *SYK, SOS1, CD79A,*
*PIK3AP1, RASGRP3*, and *MAP4K1* as DEGs in B-cells. Of note, *RASGRP3* and *PIK3AP1* did not display any expression differences between WM and IgM MGUS thus, suggesting a possible implication in the risk of progression from IgM MGUS to WM (Table 3).

### 3.5. Complement and Activation Cascade and Platelet Activation

VWF plays a crucial role in primary hemostasis by the stimulation of membrane receptors. Our study has highlighted DEGs belonging to these pathways in B-cells, as shown in Appendix A. In particular, we demonstrated that *VWF* was down regulated in WM vs. IgM MGUS according to literature [53]. Interestingly, some authors demonstrated that low levels of VWF were associated with high serum IgM levels and with *CXCR4* mutations in WM patients, and that response to treatment improves VWF in WM patients [8].

### 3.6. Cytokine-Cytokine Receptor Interactions and IL-17 Signaling Pathway

Cytokines are molecules produced by various cells and responsible for cell growth, cell death, innate and adaptive inflammatory host defenses, and processes involved in homeostasis. They induce responses through the binding to specific receptors on the surface of target cells. As a matter of fact, WM cells are in close contact with BM microenvironment composed of T cells, mast cells, monocytes, and macrophages which promote the production of cytokines and chemiokines, thus promoting malignant cell proliferation [54].

PF4 and PF4V1 chemiokines bind to CXCR3 receptor and are released during platelet aggregation playing a strong antiangiogenic function. *PPBP* encodes a Pro-Platelet Basic Protein which binds to CXCR2 receptor (KEGG). We have seen the over expression of *PF4, PF4V*, and *PPBP* in IgM MGUS and CTRLs vs. WM with high FC, whereas no gene expression differences were pointed out between IgM MGUS and CTRLs, suggesting that the down regulation of these genes was specific for WM B-cells. We could suppose a possible role of these genes in platelet dysfunction in WM because of an interaction between the IgM paraproteins and platelet surface membrane glycoproteins, which can result in the prolongation of the bleeding time and a decrease in platelet adhesion.

Finally, IL17RB takes part in the IL17 signaling pathway as well as in the Cytokine-cytokine receptor interaction pathway. IL17B and IL17E molecules bind to IL17RB in order to control the growth and differentiation of hematopoietic cells through the activation of NF-kappa B signaling pathway. Hunter et al. demonstrated the up regulation of *IL17RB* in WM patients with MYD88^L265P^ CXCR4^WT^ compared to healthy donors and to WM patients carrying different genotypes [55]. Accordingly, our GEP data showed the over expression of *IL17RB* in WM vs. IgM MGUS and vs. CTRLs with high FC.

### 3.7. Gene Expression Alterations of Signaling Pathway Responsible for Cell Cycle, Proliferation and Survival (JAK/STAT, FOXO, TGF-β, NFKB, and p53 Signaling Pathways)

The deregulation of JAK/STAT signaling pathway has been largely investigated in the pathogenesis of WM. The JAK/STAT pathway controls several cell processes through signal cascades from the cytoplasm to the nucleus. The cytokine–cytokine receptor interactions, as previously shown, include the binding between growth factors with membrane receptors (IL4R, IL21R, and IL13RA1), which activate STATs genes (i.e., *STAT3*) through JAK family genes.

Several cytokines promote the malignant cell growth and participate in the maintenance of high levels of IgM in the tumor microenvironment [54].

*BCL2* plays a crucial role in the survival of WM B-cells and the activation of *STAT3* has been shown in WM cells [51,56].

A previous study demonstrated that *IL4R* was under expressed in WM B-cells compared to normal B-cells [42]. In addition, Gutierrez et al. showed that both *IL4R* and *IL13RA1* were under expressed in WM B-cells and that these genes were not responsible for the survival of B-cells in WM [15]. On the other hand, Sekiguchi et al. demonstrated that *IL21R* over expression was associated with 6q del in WM [48].

Our GEP results correlate with previous data, suggesting that WM B-cells displayed a different JAK/STAT signature compared to IgM MGUS B-cells. Of note, all the DEGs did not show expression differences between IgM MGUS and CTRLs suggesting that JAK/STAT signaling pathway is activated in WM B-cells.

Previous data demonstrated several functions of FOXOs in cancer [49,57]. The role of IL6 in WM is very well defined [58]. As a matter of fact, the presence of IL6 in the tumor microenvironment stimulates WM cell growth and the secretion of IgM protein [54]. Studies in SCID mice implanted with WM cell lines which were treated with an anti-IL6R antibody, showed lower levels of IgM and a reduction of tumor growth [58]. Within the FOXO signaling pathway, IL6 activates STAT3 which phosphorilates the transcription factor FOXO3. This last enters in the nucleus and activates genes involved in cell cycle, apoptosis, metabolism, regulation of autophagy, and immuno-regulation (KEGG pathway hsa04068). An immunohistochemical study analyzed the presence of FOXO protein in some subpopulations of the WM cell line MWCL [59]. The authors demonstrated the presence of FOXO3a in the nuclei of CD20−CD138− cells and in the cytoplasm of CD20+CD138− and CD20+CD138+ cells.

Among the genes activated by *FOXO*, *CDKN1B* regulates the cell cycle progression in G1phase, whereas *RAG1* controls the activation of immunoglobulin V-D-J recombination. *RAG1* was found to be significantly over expressed in WM patients with respect to healthy donors [55].

A previous study described the relationship between *MYD88* and *SMADs* genes in WM. Indeed, the MYD88 protein directly interacts with SMAD6 which negatively regulates the TGF-β family signaling pathway [60].

Our GEP data highlighted that eight genes (*FOXO3, IL6, CDKNB1*, *EP300, RAG1, TGFB2, SMAD3*, and *SMAD6*) were significantly differentially expressed among WM, IgM MGUS and CTRLs B-cells. In line with previous studies, *IL6* and *FOXO3* were up regulated in WM vs. IgM MGUS. Conversely to a previous study [55], we found that *RAG1* was progressively over expressed in WM vs. IgM MGUS vs. CTRLs and we will further investigate this gene to clarify the disparity shown by our data.

Some studies demonstrated the implication of *NFKB* genes in survival, growth and resistance to treatment of B-cells in WM [10,49]. In particular, chromosome 6q deletions and *MYD88* mutations promote the activation of prosurvival signaling pathway NFKB in WM B-cells [7]. Moreover, *NFKBIZ* mutations which activate NFKB signaling pathway have been found in WM B-cells [7,10]. We have demonstrated that *NFKBIZ* showed the same expression level in IgM MGUS and CTRLs, suggesting that its over expression was specific for WM B-cells.

Akyurek et al. demonstrated the over expression of *XIAP* in cell lines of NHL and Hodgkin Lymphoma (HL) as a mechanism of apoptosis resistance [61]. According to these results, our GEP data showed the up regulation of *XIAP* in WM vs. IgM MGUS and vs. CTRLs B-cells.

### 3.8. Deregulation of p53 Signaling Pathway

P53 signaling pathway is activated by DNA damage, cellular stress, oncogenes, and oxidative processes. We have demonstrated *TP73, CD82* and *CHEK1*, as significant DEGs in B-cells.

In turn, P53 regulates target genes devolved to apoptosis, cellular senescence and cell cycle arrest. *TP73* activation works with a p53 negative feedback whereas *CD82* is involved in angiogenesis. *CD82* was over expressed in tissue samples of Nodal marginal zone lymphoma patients (NMZL) [62]. *CHEK1* (checkpoint kinase 1) is required for checkpoint mediated cell cycle arrest in response to DNA damage or the presence of unreplicated DNA. We have demonstrated that *CHEK1* did not show expression differences between WM and IgM MGUS; thus, we could speculate that this gene could be involved in the risk of progression of IgM MGUS to WM (Table 3). Our results correlate with previous studies, which have shown the use of CHEK1 inhibitors in leukemias [63,64].

### 3.9. Deregulation of Genes in WM vs. IgM MGUS vs. CTRLs in Plasma Cells

Plasma cells represent part of WM tumor cell population. A gene expression profiling study compared WM plasma cells to MM and normal CD138+ cells. The authors identified different expression signatures in plasma cells of WM with respect to MM [15]. Pals et al. demonstrated the involvement of chemokine adhesion and migration in B-cells and the homing of malignant plasma cells in tumor microenvironment in B-cell malignancies [65]. In particular, high *TJP1* expression in myeloma plasma cells was associated with a better and longer response to protease inhibitors (PI), suggesting this gene as a biomarker in order to identify patients who can benefit from PI-based therapy [66]. Our GEP results found *TJP1* as progressively over expressed from WM to IgM MGUS to CTRLs with high FC, suggesting its possible role in WM and in IgM MGUS plasma cells which need to be further investigated. As previously shown, we found only very few significant DEGs in the comparison among WM vs. IgM MGUS vs. CTRLs plasma cells, suggesting that this cell population could play a minor role compared to B-cells. However, plasma cells showed the down regulation of genes and pathways mostly involved in cell adhesion, gap and tight junctions in WM vs. IgM MGUS vs. CTRLs with very high FC.

## 4. Materials and Methods

### 4.1. Patients

We collected samples from consenting patients with WM, IgM MGUS, and healthy subjects and written informed consent was obtained from all subjects involved in the study. Among WM patients, 15/36 were asymptomatic, whereas 21/36 were symptomatic. No relapsed WM cases were included in the study. We classified WM and IgM MGUS patients according to the consensus panel recommendations from the second International Workshop on Waldenstrom’s Macroglobulinemia (IWWM) [67]. Regarding the bone marrow infiltration, patients with unequivocal BM infiltration by lymphoplasmacytic lymphoma were classified WM. While patients with no evidence of BM infiltration were considered IgM MGUS. We also classified as IgM MGUS those patients with equivocal BM infiltrate without confirming phenotypic study [67]. The study was approved by the Ethics Committee ASST Grande Ospedale Metropolitano Niguarda (Milan, Italy) with the number 195-2010-009 on 16 February 2010, according to the Declaration of Helsinki.

We enrolled 36 patients with WM, 13 subjects with IgM MGUS, and 7 healthy subjects used as CTRLs. Among them, BM CD19+ cells (*n* = 36) and BM CD138+ (*n* = 32) cells were isolated from WM, BM CD19+ cells (*n* = 13) and BM CD138+ (*n* = 10) cells were selected from IgM MGUS while BM CD19+ cells (*n* = 7) and BM CD138+ (*n* = 7) cells were isolated from CTRLs. Baseline patients’ characteristics are shown in Table 6.

### 4.2. Isolation of BM CD19+ Cells and BM CD138+ Cells Using Immunomagnetic Beads

We collected samples from all 56 subjects and we isolated mononuclear cells (MNCs) from the BM blood samples (range, 4–6 mL) using Ficoll density gradient centrifugation at 800 rpm for 20 min. Immediately after, we selected BM CD19+ cells using Human CD19 MicroBeads; afterward, CD138 cells were positively isolated from the collected CD19− cells using Human CD138 Micro Beads according to the manufacturer’s instructions (Miltenyi Biotec, Milan, Italy) [17].

### 4.3. Cell Cryopreservation and RNA Extraction

Selected BM CD19+ and BM CD138+ cells were resuspended in 50 μL of RNAlater (Thermo Fisher Scientific, Milan, Italy) and stored at −80 °C until RNA extraction was performed as previously described [17].

Total RNA was isolated from the BM CD19+ and BM CD138+ cells stored in RNAlater using MagMAX 96 Total RNA Isolation Kit (Thermo Fisher Scientific), according to the manufacturer’s instructions. The quality and the yield of the extracted RNA were measured using Nanodrop (Thermo Fisher Scientific).

### 4.4. GEP Experiments

We performed microarray experiments on the BM CD19+ and CD138+ cells. cDNA was prepared starting from the previously extracted RNA (50 ng) using Ovation Pico WTA System V2 kit (TECAN, Leek, The Netherlands) and Encore Biotin Module Kit (NuGEN) following the manufacturer’s instructions (see http://dx.doi.org/10.17504/protocols.io.yncfvaw, 19 July 2019). cDNA was hybridized to Affymetrix Human Genome U133 plus 2.0 array using the Gene Chip platform (Affymetrix, Santa Clara, CA, USA) and signals were scanned by Affymetrix Gene Chip Scanner 3000 according to the manufacturer’s instructions as described in our previous manuscript [17].

### 4.5. Gene Expression Data Analyses

Data were preprocessed and normalized using RMA and ComBat. Selection of differentially expressed genes was performed separately for CD19+ and CD138+ cells, using SAM on the 3 groups and a false discovery rate threshold of 5%, followed, for significance comparisons, by a pair-wise SAM test corrected for multiple testing (3 tests, Bonferroni correction on significance level alpha = 5%). The boxplots shown in Figure 1, were used for graphically depicting the observed expression values of the 9 genes, which indicate a possible risk of progression of IgM MGUS to WM in B-cells using R default parameters for boxplot function.

The MST in Figure 2 was computed by calculating the Euclidean pair-wise distance among subjects based on the same 9 genes. Using this distance as weight, MST is constructed as a weighted undirected graph that connects all the subjects together without any cycle and with the minimum possible total edge weight. In other word, a MST connects all the nodes (the subjects) with the minimum total sum of distances (edges in the graph) among subjects. All the analysis and figures were performed using R statistical software v. 3.6.3 (https://cran.r-project.org/, 29 February 2020).

## 5. Conclusions

In conclusion, we show that comparative transcriptome profiles of WM vs. IgM MGUS vs. CTRLs allowed to highlight genes and pathways deregulated in B-cells and in plasma cells. Interestingly, we show a common small gene set signature which characterizes WM and IgM MGUS B-cells compared to CTRL B-cells. Based on this gene set signatures, no differences in gene expression were observed between WM and IgM MGUS patients, suggesting new candidate markers in IgM MGUS subjects for the risk of progression to WM.

## Figures and Tables

**Figure 1 cancers-13-01837-f001:**
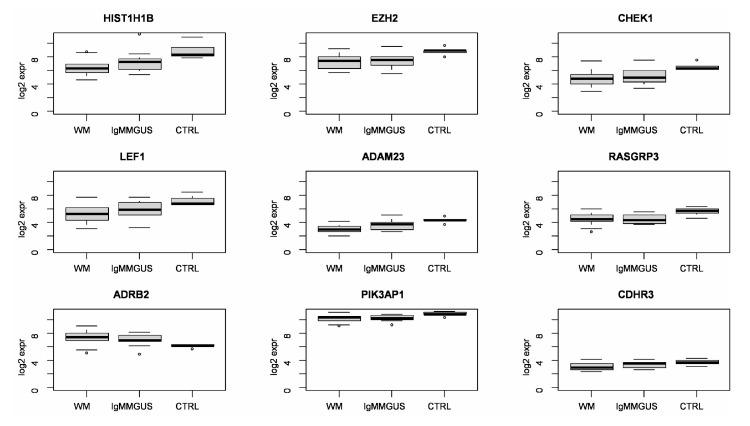
Boxplot of log base 2 expression of differentially expressed genes indicating a possible risk of progression of IgM MGUS to WM in B-cells.

**Figure 2 cancers-13-01837-f002:**
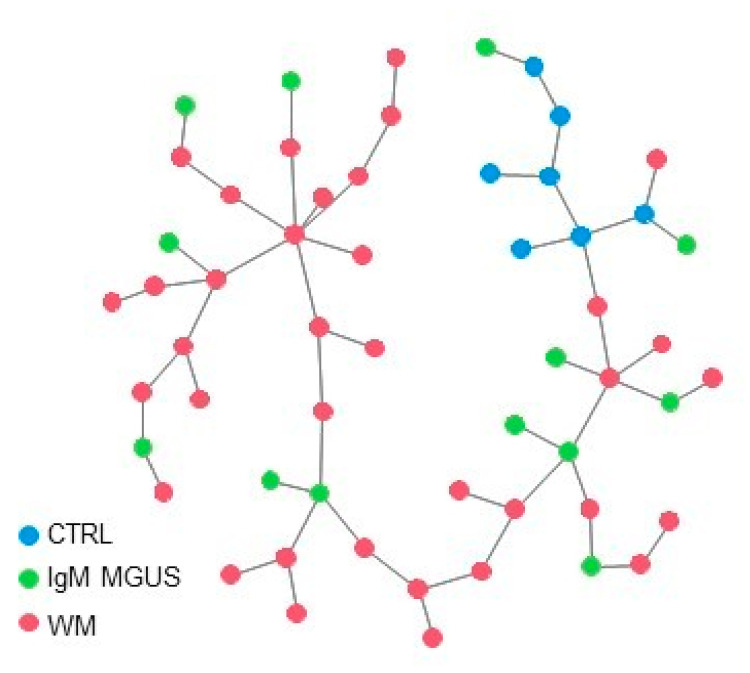
MST connecting the 3 groups of subjects based on the distance between the expression levels of the 9 selected genes. Blue, green and red dots indicate CTRL, IgM MGUS and WM samples, respectively.

**Table 1 cancers-13-01837-t001:** ZNF differentially expressed in CD19+ cells.

ZNF Differentially Expressed in CD19+ Cells								
Gene Symbol	Pattern	q Value	Mean WM	Mean MGUS	Mean CTRL	FC (*p* Value) WM vs. MGUS	FC (*p* Value) MGUS vs. CTRL	FC (*p* Value) WM vs. CTRL
ZNF804A	p1	0.0021	8.36	6.01	5.12	5.12 (0.00036)	1.85 (0.184)	9.48 (0.000003)
ZNF215	p1	0.0273	3.44	2.85	2.60	1.51 (0.000202)	1.19 (0.14)	1.79 (0.000784)
IKZF2	p2	0.0453	6.49	7.72	8.83	−2.33 (0.025399)	−2.16 (0.026)	−5.05 (0.000026)

Gene symbol is reported together with the pattern of expression (p1-p4), the q-value of the SAM statistics across the three groups, the average expression in WM, IgM MGUS and CTRL groups, and the pair-wise FC (*p*-value of the SAM statistics between the two groups in parenthesis).

**Table 2 cancers-13-01837-t002:** Differentially expressed genes in pattern 2 (CD19+) with high FC in WM vs. CTRL.

Genes Differentially Expressed in Pattern 2 CD19+ with Very High FC								
Gene Symbol	Pattern	q Value	Mean WM	Mean MGUS	Mean CTRL	FC (*p* Value) WM vs. MGUS	FC (*p* Value) MGUS vs. CTRL	FC (*p* Value) WM vs. CTRL
HIST1H1B	p2	0.0093	6.41	7.18	8.85	−1.71 (0.118)	−3.18 (0.022)	−5.45 (0.000477)
ADARB1	p2	0.0127	5.98	7.11	8.21	−2.19 (0.011)	−2.16 (0.015)	−4.72 (0.000004)
ADAM28	p2	0.0171	5.71	6.76	7.44	−2.08 (0.008)	−1.6 (0.154)	−3.33 (0.000568)
APBB2	p2	0.0248	4.61	5.88	7.07	−2.42 (0.024)	−2.27 (0.034)	−5.5 (0.000009)
EZH2	p2	0.0374	7.25	7.37	8.79	−1.09 (0.733)	−2.67 (0.002)	−2.91 (0.000034)

Gene symbol is reported together with the pattern of expression (p1–p4), the q-value of the SAM statistics across the three groups, the average expression in WM, IgM MGUS and CTRL groups, and the pair-wise FC (*p*-value of the SAM statistics between the two groups in parenthesis).

**Table 3 cancers-13-01837-t003:** Differentially expressed genes indicating a possible risk of progression of IgM MGUS to WM in B-cells.

DEGs Indicating a Possible Risk of Progression of IgM MGUS to WM in B Cells								
Gene Symbol	Pattern	q Value	Mean WM	Mean MGUS	Mean CTRL	FC (*p* Value) WM vs. MGUS	FC (*p* Value) MGUS vs. CTRL	FC (*p* Value) WM vs. CTRL
HIST1H1B	p2	0.009	6.41	7.18	8.85	−1.71 (0.118)	−3.18 (0.022)	−5.45 (0.000477)
EZH2	p2	0.037	7.25	7.37	8.79	−1.09 (0.733)	−2.67 (0.002)	−2.91 (0.000034)
CHEK1	p2	0.040	4.81	5.14	6.49	−1.26 (0.411)	−2.55 (0.004)	−3.22 (0.000024)
LEF1	p2	0.041	5.24	5.82	7.19	−1.5 (0.19)	−2.58 (0.01)	−3.86 (0.000136)
ADAM23	p2	0.045	4.12	4.84	6.15	−1.64 (0.137)	−2.48 (0.011)	−4.07 (0.000032)
RASGRP3	p3	0.045	4.57	4.47	5.60	1.07 (0.687)	−2.19 (0.001)	−2.05 (0.002118)
ADRB2	p1	0.047	7.33	6.98	6.13	1.28 (0.226)	1.8 (0.005)	2.31 (0.000003)
PIK3AP1	p2	0.047	10.15	10.24	10.84	−1.06 (0.565)	−1.51 (0.002)	−1.61 (0.000206)
CDHR3	p2	0.048	4.82	5.31	6.29	−1.4 (0.142)	−1.98 (0.028)	−2.77 (0.001467)

Gene symbol is reported together with the pattern of expression (p1–p4), the q-value of the Statistical Analysis for Microarrays (SAM) statistics across the three groups, the average expression in WM, IgM MGUS and CTRL groups, and the pair-wise Fold Changes (FC) (*p*-value of the SAM statistics between the two groups in parenthesis).

**Table 4 cancers-13-01837-t004:** Differentially expressed genes in CD138+ cells.

Genes Differentially Expressed in CD138+								
Gene Symbol	Pattern	q Value	Mean WM	Mean MGUS	Mean CTRL	FC (*p* Value) WM vs. MGUS	FC (*p* Value) MGUS vs. CTRL	FC (*p* Value) WM vs. CTRL
TJP1	p2	0.0005	3.62	5.02	6.24	−2.65 (0.002)	−2.33 (0.013)	−6.18 (0.0000003)
ESRP1	p2	0.002	4.44	5.4	6.44	−1.94 (0.002)	−2.07 (0.008)	−4.02 (0.000005)
FGF2	p2	0.0021	2.25	2.45	3.26	−1.15 (0.072)	−1.75 (0.005)	−2.01 (0.000822)
SEPT10	p2	0.0082	3.93	5.02	6.02	−2.14 (0.004)	−1.99 (0.025)	−4.26 (0.000007)
GREB1	p3	0.0092	1.92	1.75	3.46	1.12 (0.005)	−3.28 (0.005)	−2.92 (0.013437)
PERP	p2	0.0097	3.39	4.28	5.51	−1.85 (0.03)	−2.35 (0.002)	−4.4 (0.000000002)
NFIB	p2	0.0134	2.46	3.38	4.06	−1.9 (0.022)	−1.59 (0.138)	−3.04 (0.000078)
CALCRL	p2	0.0164	3.05	3.87	4.36	−1.77 (0.008)	−1.41 (0.171)	−2.48 (0.000142)
PPARGC1A	p2	0.0191	3.32	4.49	4.49	−2.24 (0.004)	−1 (0.993)	−2.25 (0.000901)
TUBB2B	p2	0.0225	3.26	3.31	4.85	−1.04 (0.729)	−2.92 (0.006)	−3.03 (0.0065)
PLA2G2D	p2	0.0276	4.85	5.6	6.62	−1.69 (0.019)	−2.02 (0.003)	−3.41 (0.0000004)
PARD3	p2	0.0279	5.23	5.95	6.56	−1.65 (0.005)	−1.52 (0.071)	−2.52 (0.000226)
OVOL1	p2	0.0293	5.49	6.03	6.54	−1.46 (0.016)	−1.42 (0.059)	−2.07 (0.000058)
TTLL7	p2	0.0341	2.92	3.18	4.07	−1.2 (0.175)	−1.86 (0.011)	−2.22 (0.001204)
GBA3	p4	0.041	3.43	5.1	4.58	−3.2 (0.004)	1.44 (0.316)	−2.22 (0.001208)
MYO5B	p2	0.0422	5.03	5.66	6.37	−1.55 (0.025)	−1.63 (0.043)	−2.52 (0.000136)
SYNM	p2	0.0426	4.84	5.81	5.91	−1.97 (0.003)	−1.07 (0.75)	−2.11 (0.000054)
KIAA1804	p2	0.0428	3.92	4.28	5.28	−1.29 (0.216)	−1.99 (0.003)	−2.57 (0.0000005)
NRG3	p4	0.043	3.01	4.05	4.03	−2.06 (0.002)	1.01 (0.959)	−2.03 (0.007143)
KCTD1	p2	0.0434	5.27	5.77	6.35	−1.41 (0.024)	−1.5 (0.018)	−2.11 (0.000012)
CNTN1	p2	0.0435	2.56	3.03	3.39	−1.39 (0.011)	−1.29 (0.163)	−1.79 (0.00149)
LAPTM4B	p2	0.0437	4.71	5.27	6.49	−1.48 (0.086)	−2.32 (0.001)	−3.42 (0.000001)
CADPS2	p2	0.0438	5.47	6.87	8.12	−2.63 (0.011)	−2.38 (0.014)	−6.26 (0.0000005)
ARHGAP32	p2	0.0451	4.02	4.85	5.4	−1.77 (0.002)	−1.47 (0.061)	−2.61 (0.000052)
GPR125	p2	0.0453	3.58	4.05	4.93	−1.38 (0.031)	−1.84 (0.012)	−2.55 (0.000346)
UCHL1	p2	0.0454	5.24	5.86	6.52	−1.54 (0.065)	−1.58 (0.051)	−2.42 (0.000002)
MARC2	p2	0.0461	4.8	6.18	6.46	−2.59 (0.001)	−1.22 (0.433)	−3.17 (0.000017)
TTLL7	p2	0.047	4.61	5.45	6.27	−1.79 (0.03)	−1.76 (0.055)	−3.15 (0.00004)
MYLK	p2	0.0486	6.13	6.96	7.53	−1.78 (0.007)	−1.49 (0.072)	−2.64 (0.000028)

Gene symbol is reported together with the pattern of expression (p1–p4), the q-value of the SAM statistics across the three groups, the average expression in WM, IgM MGUS and CTRL groups, and the pair-wise FC (*p*-value of the SAM statistics between the two groups in parenthesis).

**Table 5 cancers-13-01837-t005:** Common DEGs in CD19+ cells and CD138+ cells ordered by pattern and minimum fold change across different comparison.

Gene Symbol	Pattern CD19+	Pattern CD138+	CD19+			CD138+		
			FC WM vs. MGUS	FC MGUS vs. CTRL	FC WM vs. CTRL	FC WM vs. MGUS	FC MGUS vs. CTRL	FC WM vs. CTRL
ARHGAP32	p2	p2	−2.52	−1.55	−3.9	−1.77	−1.47	−2.61
MARC2	p2	p2	−2.47	−1.51	−3.72	−2.59	−1.22	−3.17
LAPTM4B	p2	p2	−1.76	−1.57	−2.76	−1.48	−2.32	−3.42
PLA2G2D	p2	p2	−1.5	−1.1	−1.65	−1.69	−2.02	−3.41
KIAA1804	p2	p2	−1.36	−2.02	−2.76	−1.29	−1.99	−2.57
MYLK	p2	p2	−2.04	−1.28	−2.61	−1.78	−1.49	−2.64
SYNM	p2	p2	−1.59	−1.15	−1.84	−1.97	−1.07	−2.11
TJP1	p4	p2	−1.33	1.09	−1.22	−2.65	−2.33	−6.18
GBA3	p4	p2	−1.87	1.51	−1.24	−3.54	1.44	−2.46
NFIB	p4	p2	−1.77	1.12	−1.59	−1.9	−1.59	−3.04
PPARGC1A	p4	p2	−1.27	1.15	−1.1	−2.24	−1	−2.25

Gene symbol is reported together with the pattern of expression (p1–p4) in CD19+ and CD38+ cells, together with the pair-wise FC between groups in the two cell types.

**Table 6 cancers-13-01837-t006:** Clinical characteristics of WM and IgM MGUS patients.

Parameter	Total *n* = 49	WM *n* = 36	IgM MGUS *n* = 13
Median age (years)	68	69	67
Sex *n* (%)			
Male	20 (42)	14 (39)	6 (50)
Female	29 (58)	22 (61)	7 (50)
% of bone marrow involvement (median)	20	30	2
M-protein level (g/dL) median	0.8	1.1	0.3
Light chain			
κ *n* (%)	35 (71)	25 (69)	10 (77)
λ *n* (%)	9 (18)	7 (19)	2 (15)
Both *n* (%)	3 (6)	3 (8)	0 (0)
Missing data n	2	1	1
Haemoglobin (g/dL) median	12.2	11.3	12.9
Beta2 microglobulin (μg/mL) median (interval)	2.9 (2–8.24)	3.1 (1.87–8.24)	2.3 (2–7.5)
Lactate dehydrogenase (U/L, UL ^1^ 480) median (interval)	200 (85–670)	200 (85–670)	200 (156–325)
Presence of BJ ^2^ proteinuria *n* (%)	19 (39)	18 (50)	1 (8)
Presence of immunoparesis *n* (%)	10 (20)	10 (28)	0 (0)

^1^ Upper Laboratory Limit; ^2^ Bence Jones.

## Data Availability

The data presented in this study are openly available in Gene Expression Omnibus repository at https://www.ncbi.nlm.nih.gov/geo/ with the ID GSE171739.

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
