# Peer review of "Identification of a Candidate Gene Set Signature for the Risk of Progression in IgM MGUS to Smoldering/Symptomatic Waldenström Macroglobulinemia (WM) by a Comparative Transcriptome Analysis of B Cells and Plasma Cells"

_cancers, 2021, doi:10.3390/cancers13081837_

Round 1

Reviewer 1 Report

Manuscript Review

The manuscript titled "Identification of a candidate gene set signature for the risk of progression in IgM monoclonal gammopathy of undetermined significance (IgM MGUS) to Waldenström Macroglobulinemia (WM) by a comparative transcriptome analysis of B cells and plasma cells"  aims to determine unique gene signatures for progression of IgM MGUS to WM. To this end, the authors performed a gene expression profiling (GEP) study to compare the transcriptome signatures of bone marrow B cells (CD19+) and plasma cells (CD138+) of 36 WM and 13 IgM MGUS patients and 7 healthy donors (CTRL). Briefly, the authors identified 2038 unique genes for the CD19+ population and 29 unique genes in the CD138+ population. They clustered the GEPs in 4 groups, with the difference in each gene expression level in the specific sample: highest expression in WM samples, highest expression in IgM MGUS samples, highest expression in CTRL samples, high expression in WM, and CTRL samples. Next, they performed a function annotation clustering and enrichment analysis for each of those groups using DAVID 6.8 (https://david.ncifcrf.gov/). Finally, the authors discuss their results in the context of the individual pathways and compare them to published literature.

The research topic and aims are important and would be very useful in predicting the progression risk from IgM MGUS to WM. However, there are two significant concerns with this study. First, the novelty of this manuscript is disputable at best. The nine identified candidate genes for the risk of progression from IgM MGUS to WM could not reach the transcriptomic study's primary goal (i.e., provide novel information on the gene set signature for the risk of progression from MGUS to WM). Second, the overall quality manuscript (data presentation, clarity, format, language, style – see specific comments below) feels sloppy and does not meet the journal's standards. Therefore, the manuscript cannot be recommended for publication in its current form.

Major comments:

Conclusions:  The authors conclude that the study "doesn't bring any new information on the gene set signature for the risk of progression from MGUS to WM" but claim without any further explanation that "no differences are suggesting possible mechanisms of leukemogenesis." Statements like that should be explained and reformulated for clarity.

Data analysis: The nine candidate genes have similar GEPs between WM and IgM MGUS and some differences compared to CTRL samples (Table1). I can agree with the comparable expression levels of most of them, but for PIK3AP1, for example, the fold change for MGUSvsCTRL and WMvsCTRL falls into the range which the authors consider as "no significant change" for WMvsMGUS. Based on the presented data, maybe six out of nine genes can be considered significantly differentially expressed. A plot would help visualize the data to understand better what is significant and what is not.

Data presentation: In general, the data should be presented more clearly, e.g., the numbers of unique genes for specific patterns in paragraph 2.3. Rather than continually referring to the same supplementary table, I would suggest a summary table for each paragraph. A graphical representation would also help visualize the expression of selected genes (perhaps a box plot with error bars for a clearer understanding of the expression profile rather than just the mean value for each sample group).

Tables: The data in Table 1 (and other tables) has been done without any sorting (not alphabetically for gene symbols nor in ascending order for a specific column). It seems like the authors pasted the data in the tables randomly without any criteria.

Figures: The minimum spanning tree does not provide any useful information. It should be better explained or deleted entirely.

4. Messed-up supplementary tables: Throughout the manuscript, the authors refer to supplementary Table S1 and S1-1, which is not provided. In the supplement, only 3 files are available: PatternsCD19, PatternsCD138 and commonDE_annot. Moreover, all supplementary table titles are written in Italian (using prepositions like "con," "ma," "TRA," "e," and "NESSUNO").

Minor comments:

Language and grammar: The manuscript needs language editing, and all grammatical errors should be corrected: e.g., "significantly differently expressed" instead of "differentially," use of prepositions (for example, row 64 "from asymptomatic WM (smoldering/smWM) form to symptomatic WM" or row 183 "progression in IgM MGUS to WM").

Reviewer 2 Report

Trojani et al have produced an impressive transcriptional characterization of the full breadth of WM disease including WM and IgM MGUS as well as B-cell and Plasma cell controls. Putting aside the bioinformatic analysis for a minute, just the production this kind of comprehensive data set in a rare disease is an achievement in and of itself for which the authors are to be congratulated. The most important aspect of this manuscript is identification or nine genes that are similar in expression between WM and IgM MGUS but dysregulated relative to healthy donor controls. The genes in this list are of biological relevance and could add to our understanding of IgM MGUS and progression to full WM. That being said, the paper is not without issues, including concerns with how the samples were generated, lack of MYD88 and CXCR4 genotype information and insufficient bioinformatic engagement with the central finding. While the paper contains many interesting insights, even in the discussion it is structured more like a series of result tables and is not ready for publication without revision.

Major Concerns:

  • The definition of IgM MGUS is quite controversial. Some experts like Steven Treon argue that any meaningful LPC involvement in the bone marrow is an indicator or sWM if not full-blown disease while Robert Kyle has long argued for minimum LPC involvement levels for each stage. There is no definitive answer at this time certainly nothing wrong with the author’s apparent approach. However, because the definition is not agreed upon, it is critical that the authors fully document how the IgM MGUS samples were identified and what criteria were used to distinguish them from WM. As the inclusion of IgM MGUS with the other samples is the major contribution of the paper, this must be stated very clearly in the methods and engaged thoroughly in the discussion.
  • On a similar note, what efforts were undertaken measure the purity of the MGUS samples? In IgM MGUS, CD19 selection alone may lead to the inclusion of a significant quantity of regular non-clonal CD19+ B-cells. In extreme cases the GEP signal may be how regular B-cells respond to BM environment generated by IgM MGUS. Likewise, the timing on MGUS is equally complicated. If MGUS has actually transformed to full WM it may take a significant amount of time to grow out, during which it still may meet the authors definition of MGUS. I do not raise these issues to invalidate the findings, but these issues are important and should be addressed in the discussion.
  • Is any CXCR4 or MYD88 genotyping available for these samples? In particular many MYD88 wild type WM patients have extremely low levels of BM infiltration in spite of having advanced, aggressive disease which may lead to their inclusion in IgM MGUS. While this is more of a theoretical concerns given how rare this WM subpopulation is, the larger issue is the extreme degree of heterogeneity observed in WM based on MYD88 and CXCR4 mutation status. As mentioned by the authors, genes like IL17RB have an almost bimodal expression pattern based on CXCR4 status. Taking the mean of the whole population dilutes the finding and leads high levels of variance in the WM data. If such genotype data is available, including it in the DGE modeling, even if it is not used to generate genotype specific results, may lead to significant improvements DEG identification just be accounting for this source of variance in the data.
  • The 9-gene list is just presented, analyzed with no additional follow up or study. This is perhaps the most interesting finding and it seems strange no additional work was done. Do the authors have access to a cohort of IgM MGUS samples that ultimately transformed in WM or other NHL? Could they be tested for these 9 genes? How about expression levels vs. time from diagnosis to transformation to WM? If there is really nothing that can be added here, could the authors suggest studies in the discussion that would help address their hypothesis? How might these genes be used to drive WM research?
  • The spanning tree is insufficient. It’s the only figure yet has no meaningful discussion about methodology or what it is showing which is important as many readers are likely unfamiliar with this approach. Even so, the tree itself is not supper convincing for the central argument unless the sole point is that the CNTRL samples form a cluster. Given that the genes were IDed by being DEG relative to the CTRL samples, this is not a surprise. Were other approaches considered as alternatives or in addition to this tree? Off the top of my head I would at least consider MDS/PCA, tSNE, UMAP, or hierarchical clustering, though I am sure there are other, better approaches. At the very least, please add methodology and add more discussion of the figure and what it demonstrates.
  • Finally, both the results and the discussion seem to be a straight read out of results. Some of this is unavoidable in this sort of study, but it makes the discussion feel a bit repetitive where it should be synthesizing and showing how these results work together as a whole. Condensing the results to make room for a more integrated in comprehensive discussion may be beneficial, though this is obviously a matter of opinion.

Minor Comments:

  • While CD79A mutations have been reported in WM several studies have noted that these are relatively rare when compared with the rate of CD79B mutations. Possibly a typo?
  • The RAG1 findings in this study certainly frankly make more sense as one might expect the RAGs to be downregulated after VDJ recombination. While GEP data from Chng et al and Hunter et al do seem to show the opposite, your data is what it is. I very much appreciated that you flagged the differences for further study.
  • Perhaps I am not reading this correctly but would not a smaller number of differences between WM and CTRL plasma cells indicate increased similarity compared to the results from CRTL B-cells? Can this be clarified?

Reviewer 3 Report

Waldenström's macroglobulinaemia (WM) is a peculiar disease characterized by lymphoplasmocytic lymphoma predominanly occupying the bone marrow and production of IgM paraprotein. It is accepted that that the disease biologically is somewhat closer to CLL than to multiple myeloma.

In this groundbreaking study the authors attempt to use transcriptome analysis in order to delineate the process of transformation of IgM MGUS to manifest WM. The paper is presented in rich detail and is of potential great importance to workers of the fied of WM. The sample size of 36 WM patients is significant in this rare disease. Control sample size of IgM MGUS on the other hand is somewhat low of only 13 individuals. And there are some irregularities of that could be discerned on Table 6. First and foremost, the amount of IgM paraprotein in WM is rather low, one may expect mean M-protein size for newly diagnosed WM in the range of 30 g/l (about 3 time higher than observed in this study). Were all patients newly diagnosed and symptomatic with this low M-protein amount? Or did the authors include samples from relapsed WM cases? This needs clarification as it may greatly influence the conclusuions of the manuscript. Additionally, 3% of immunoparesis (definition lacking) is unexpectedly low, >50% would be expected in this elderly patient population of indolent lymphoma. Similarly, median hemoglobin values of near normal level are most unusual for WM. What was the definition of symptomatic WM for inclusuion in this study? Clarification and range provision would be minimally needed for these parameters. Applicability of the results could be jeopardized. Additionally, one may expect a few (1-2) IgM myeloma to emerge in a study of this site. How were these cases excluded?

Round 2

Reviewer 1 Report

Thank you for addressing most of my comments and suggestions to improve the manuscript. The whole document reads much better in the revised form, and the results are presented more clearly now. However, the updated figures with boxplots would further benefit from the same X-axis for each plot. This modification would help to evaluate differential gene expression levels at first glance. Overall, the paper can be considered for publication.

Author Response

We have changed the boxplots (Figure 1) as you have suggested (we have included the new figure immediately after line 349).

Reviewer 2 Report

The paper is much improved. For the record, I am not surprised that the spanning tree is associated with a PCA like the one attached in the response. In my experience they are telling similar stories, however my concern was about the context provided to readers who may not be familiar with this approach and this has been much improved. All my concerns have been addressed and I believe this paper will contributing meaningfully to the literature on WM.

Author Response

We would like to thank you very much for your observations and considerations.

Reviewer 3 Report

This is an improved and redesigned manuscript. From the data it is now clear that oligosymptomatic early WM contributed heavily to this study. Unfortunately, this somewhat decreases the validity of the results as the patient population is not representative for the customary WM presentation. Maybe a new title would be indicated to represent this fact. No response received to the question of excluding IgM myeloma.

Author Response

Considering your suggestion, we have changed the title of the paper as follows: Identification of a candidate gene set signature for the risk of progression in IgM monoclonal gammopathy of undetermined significance (IgM MGUS) to smoldering/symptomatic Waldenström Macroglobulinemia (WM) by a comparative transcriptome analysis of B cells and plasma cells.

In our study, we have excluded patients with suspected or confirmed morphologic and immophenotipic diagnosis of IgM myeloma (Schuster SR, Am J Hematol. 2010).